# TLR4 Stimulation Promotes Human AVIC Fibrogenic Activity through Upregulation of Neurotrophin 3 Production

**DOI:** 10.3390/ijms21041276

**Published:** 2020-02-14

**Authors:** Qingzhou Yao, Erlinda The, Lihua Ao, Yufeng Zhai, Maren K. Osterholt, David A. Fullerton, Xianzhong Meng

**Affiliations:** Department of Surgery, University of Colorado Denver, Box C-320, 12700 E 19th Avenue, Aurora, CO 80045, USA; Qingzhou.Yao@cuanschutz.edu (Q.Y.); Erlinda.The@cuanschutz.edu (E.T.); Lihua.Ao@cuanschutz.edu (L.A.); Maren.Osterholt@cuanschutz.edu (M.K.O.); David.Fullerton@cuanschutz.edu (D.A.F.)

**Keywords:** TLR4, neurotrophin 3, fibrosis, proliferation, LPS

## Abstract

Background: Calcific aortic valve disease (CAVD) is a chronic inflammatory disease that manifests as progressive valvular fibrosis and calcification. An inflammatory milieu in valvular tissue promotes fibrosis and calcification. Aortic valve interstitial cell (AVIC) proliferation and the over-production of the extracellular matrix (ECM) proteins contribute to valvular thickening. However, the mechanism underlying elevated AVIC fibrogenic activity remains unclear. Recently, we observed that AVICs from diseased aortic valves express higher levels of neurotrophin 3 (NT3) and that NT3 exerts pro-osteogenic and pro-fibrogenic effects on human AVICs. Hypothesis: Pro-inflammatory stimuli upregulate NT3 production in AVICs to promote fibrogenic activity in human aortic valves. Methods and Results: AVICs were isolated from normal human aortic valves and were treated with lipopolysaccharide (LPS, 0.20 µg/mL). LPS induced TLR4-dependent NT3 production. This effect of LPS was abolished by inhibition of the Akt and extracellular signal-regulated protein kinases 1 and 2 (ERK1/2) pathways. The stimulation of TLR4 in human AVICs with LPS resulted in a greater proliferation rate and an upregulated production of matrix metallopeptidases-9 (MMP-9) and collagen III, as well as augmented collagen deposition. Recombinant NT3 promoted AVIC proliferation in a tropomyosin receptor kinase (Trk)-dependent fashion. The neutralization of NT3 or the inhibition of Trk suppressed LPS-induced AVIC fibrogenic activity. Conclusions: The stimulation of TLR4 in human AVICs upregulates NT3 expression and promotes cell proliferation and collagen deposition. The NT3-Trk cascade plays a critical role in the TLR4-mediated elevation of fibrogenic activity in human AVICs. Upregulated NT3 production by endogenous TLR4 activators may contribute to aortic valve fibrosis associated with CAVD progression.

## 1. Introduction

Calcific aortic valve disease (CAVD), the most prevalent cardiovascular disorder in the elderly, is characterized by progressive fibrosis and the calcification of the aortic valve leaflets. Severe CAVD causes aortic stenosis that requires costly aortic valve replacement [1]. Moreover, moderate aortic valve thickening increases the risk of other cardiovascular events, such as myocardial injury and left ventricular decompensation [2,3]. Therefore, it is important to prevent CAVD progression. Unfortunately, gaps in our knowledge of the mechanism underlying CAVD progression impede the development of pharmacological therapies [4,5].

It is well known that CAVD is a chronic inflammatory condition [6], and inflammatory mediators appear to promote valvular fibrosis and calcification [7,8]. The aortic valve is composed of aortic valve interstitial cells (AVICs), an extracellular matrix (ECM), and overlying endothelial cells. AVICs are the dominant cells of valve leaflets, and are known to be important contributors to the pathogenesis of CAVD [5]. We have observed that the AVIC inflammatory response exerts an impact on cellular osteogenic activity [9]. Aortic valve fibrosis, an important pathobiology of CAVD progression, is characterized by AVIC proliferation and the excessive production and accumulation of ECM proteins, particularly collagens [5]. However, the molecular event that triggers valvular fibrosis is not fully understood.

Toll-like receptor 4 (TLR4) plays a critical role in regulating the cellular inflammatory response [9]. Our previous study found that diseased human aortic valves express higher levels of toll-like receptors (TLRs), particularly TLR2 and TLR4 [10]. These two TLRs regulate the expression of multiple pro-inflammatory mediators in human AVICs. More importantly, the stimulation of TLR4 in human AVICs induces the expression of pro-osteogenic factors to elevate valvular osteogenic activity [10]. It is unclear whether the stimulation of TLR4 induces the fibrogenic responses (cell proliferation and the over-production of the ECM protein) in human AVICs. In this regard, the prolonged activation of TLR4 by lipopolysaccharide (LPS) is found to induce kidney fibrosis [11]. In addition, TLR4 is found to play a role in the mechanism of pulmonary fibrosis following lung injury [12,13]. It is likely that TLR4 stimulation also enhances the fibrogenic activity in the aortic valve. The investigation of the impact of TLR4 stimulation on the fibrogenic activity in human AVICs may improve our understanding of the role of valvular inflammation mediated by innate immunity in the mechanism of aortic valve fibrosis associated with CAVD progression.

Neurotrophin-3 (NT3) is a member of the neural growth factor family, and it binds to the tropomyosin receptor kinase (Trk)A, TrkB, and TrkC with high affinity [14]. The main function of NT3 is to maintain the viability of existing neurons and to promote the growth and differentiation of new neurons [15]. NT3 has also been found to be an important regulator of fibroblast function, including cell migration and the secretion of cytokines [16,17]. We previously observed that NT3 is present in human aortic valves, and this growth factor is localized in AVICs and in the extracellular spaces between AVICs [18]. Interestingly, NT3 expression is enhanced in human aortic valves affected by CAVD [18]. Furthermore, NT3 is capable of promoting the osteogenic and fibrogenic activities in human AVICs [18,19]. It is likely that the over-production of NT3 plays a role in the mechanism of aortic valve fibrosis and calcification associated with CAVD progression. Currently, it is unclear whether pro-inflammatory stimuli modulate NT3 expression in the aortic valve. A previous study found that monocytes release greater levels of NT3 in response to LPS treatment [20]. It is possible that stimulation of TLR4 in human AVICs modulates NT3 production and that the TLR4-mediated mechanism contributes to elevated NT3 levels observed in diseased aortic valves.

The presented study tested the hypothesis that the stimulation of TLR4 in human AVICs upregulates NT3 production to elevate cellular fibrogenic activity. The purpose of this study was to determine: (1) the effect of the stimulation of TLR4 on the fibrogenic activity (cell proliferation, ECM protein production and collagen deposition) in human AVICs; (2) whether and how TLR4 signaling upregulates NT3 expression and (3) whether NT3 and the Trk pathway mediate the pro-fibrogenic effect of TLR4 stimulation in human AVICs.

## 2. Results

### 2.1. TLR4 Activation Induces Stronger Fibrogenic Activity in Human AVICs

TLR4 signaling upregulates the inflammatory and osteogenic activities in human AVICs and plays a mechanistic role in mediating aortic valve inflammation and calcification [10]. To determine whether stimulation of TLR4 upregulates the fibrogenic activity in human AVICs, we treated the AVICs of normal human aortic valves with LPS (0.20 µg/mL) for 72 h. Bromodeoxyuridine/5-bromo-2′-deoxyuridine (BrdU) incorporation increased by 110% in cells treated with LPS compared to untreated control cells, and the proliferative effect of TLR4 stimulation was confirmed by the cell counting kit-8 (CCK-8) assay (Figure 1A). The stimulation of TLR4 with LPS in human AVICs for 72 h also upregulated the expression of collagen III and matrix metallopeptidases-9 (MMP-9) (Figure 1B). Furthermore, the treatment of human AVICs with LPS for 28 days resulted in greater collagen deposition, an in vitro activity of fibrosis (Figure 1C). Therefore, TLR4 stimulation up-regulates the fibrogenic activity in human AVICs.

### 2.2. Stimulation of TLR4 in Human AVICs with LPS Upregulates NT3 Production

Our previous studies found that NT3 is pro-osteogenic and pro-fibrogenic in human AVICs and that levels of this neurotrophin are elevated in human aortic valves affected by CAVD [18,19]. To determine whether the stimulation of TLR4 upregulates NT3 production, we treated AVICs of normal human aortic valves with LPS (0.20 µg/mL) for 24, 48, and 72 h and applied immunoblotting to analyze the levels of NT3 protein. Figure 2A shows that the stimulation of TLR4 increases NT3 levels in human AVICs in a time-dependent fashion. At 72 h of treatment with LPS, cellular NT3 levels were doubled. In cells pretreated with a TLR4-neutralizing antibody (10 μg/mL), the effect of LPS on NT3 production was markedly reduced (Figure 2B). Thus, the activation of TLR4 by LPS induces NT3 production in human AVICs.

It should be noted that immunoblotting in a non-reducing condition detected two bands approximately 30 kDa and 65 kDa, respectively, while the molecular size of mature NT3 monomer is around 15 kDa, and the pro-form of NT3 is approximately 30 kDa. It has been documented that NT3 is present in cells as homodimer or heterodimer with a brain-derived neurotrophic factor [21,22]. Dimers of mature NT3 are approximately 30 kDa, and those pro-form should be approximately 60 kDa. The two bands we detected are NT3 in dimeric form.

### 2.3. Akt and ERK1/2 are Involved in the Mechanism by which TLR4 Activation Induces NT3 Production in Human AVICs

We have observed that the extracellular signal-regulated protein kinases 1 and 2 (ERK1/2) pathway plays an important role in the TLR4-mediated osteogenic response in human AVICs [23]. The Akt pathway was found to mediate laminin-induced NT3 production in stem cells [24]. As the stimulation of TLR4 activates multiple signaling pathways, including the Akt and ERK1/2 pathways, we determined whether the Akt and ERK1/2 pathways are involved in the TLR4-induced NT3 production in human AVICs. We treated AVICs with LPS for 1 to 8 h and examined the activation of these two signaling pathways. Figure 3A shows that LPS induces the rapid phosphorylation of both Akt and ERK1/2. The inhibition of either Akt (MK2206, 5.0 µM) or ERK1/2 (PD98059, 25 µM) markedly reduced LPS-induced NT3 production (Figure 3B). The results demonstrate that the Akt and ERK1/2 pathways are involved in the mechanism by which TLR4 stimulation induces NT3 production in human AVICs.

### 2.4. NT3 Mediates Human AVIC Proliferation Induced by TLR4 Stimulation

To determine the effect of TLR4 activation on AVIC proliferation and the role of NT3 in mediating this effect, we treated AVICs with LPS (0.20 µg/mL) for 72 h in the presence or absence of a NT3-neutralizing antibody (10 μg/mL). Cell proliferation was analyzed using a BrdU ELISA kit and a CCK-8 assay. BrdU incorporation increased by 110% in cells treated with LPS compared to the untreated control cells, and increased cell proliferation was confirmed by formazan dye formation assessed by the CCK-8 assay (Figure 4). The results in untreated control and LPS alone groups are comparable to the results of corresponding groups in Figure 1A and are presented as the combined data of multiple experiments. It is noteworthy that neutralization of NT3 essentially abrogated the cell proliferation induced by stimulation of TLR4.

Our previous studies demonstrate that Trk, a specific neurotrophin receptor, mediates NT3 action in human AVICs [18]. To confirm the mechanistic role of NT3 in mediating AVIC proliferation induced by TLR4 stimulation, we treated AVICs with LPS (0.20 µg/mL) or NT3 (0.10 µg/mL) for 72 h in the presence or absence of a specific Trk inhibitor (K252a, 0.20 μM). Figure 5A shows that inhibition of Trk abrogated the effect of NT3 on AVIC proliferation. The results of the BrdU and CCK-8 assays in Figure 5B show that the inhibition of Trk has a suppressive effect on TLR4-induced AVIC proliferation compared to that of NT3 neutralization. Together, the results demonstrated that NT3 mediates AVIC proliferation induced by TLR4 stimulation.

### 2.5. The NT3-Trk Cascade is Involved in the Mechanism by which TLR4 Stimulation Elevates Fibrogenic Activity in Human AVICs

To determine the role of the NT3-Trk cascade in elevating the fibrogenic activity (ECM protein production and collagen deposition) in human AVICs by TLR4 stimulation, we applied an NT3-neutralizing antibody (10 µg/mL) or a Trk inhibitor (K252a, 0.20 µM) to treat AVICs before exposing them to LPS. The stimulation of TLR4 with LPS for 72 h upregulated the expression of collagen III and MMP-9 (Figure 6A). The neutralization of NT3 or the inhibition of Trk markedly reduced collagen III and MMP-9 in AVICs exposed to LPS (Figure 6A). The treatment of human AVICs with LPS for 28 days resulted in greater collagen deposition, an in vitro fibrotic change (Figure 6B). The NT3-neutralizing antibody and Trk inhibitor also markedly reduced collagen deposition induced by prolonged exposure to LPS (Figure 6B). Therefore, the NT3-Trk cascade plays a major role in the mechanism by which TLR4 stimulation elevates the fibrogenic activity in human AVICs.

## 3. Discussion

CAVD progression involves valvular fibrosis and calcification, and the progressive thickening of aortic valve leaflets due to fibrosis and calcification results in valvular dysfunction. It is the scientific consensus that chronic valvular inflammation is an important factor promoting fibrosis and calcification [25]. Currently, the inflammatory mechanism of CAVD pathobiology is not well understood. The results of the present study demonstrate that the stimulation of human AVICs with a TLR4 agonist LPS upregulates cellular fibrogenic activity. Furthermore, the TLR4 agonist induces the expression of NT3 in human AVICs through activation of both Akt and ERK1/2 pathways. NT3 mediates TLR4-enhanced AVIC fibrogenic activity through the Trk receptor. These novel findings suggest that the inflammatory milieu may contribute to the mechanism underlying elevated levels of NT3 in diseased aortic valves and that NT3 may have a mechanistic role in mediating the valvular pathobiology associated with CAVD progression.

Our previous study found that NT3 is over-expressed in the aortic valves of patients with CAVD, and that NT3 promotes AVIC osteogenesis and proliferation [18,19]. The present study revealed that the stimulation of TLR4 with LPS upregulates NT3 production in human AVICs. This finding is in agreement with a previous study that demonstrated that the stimulation of monocytes with LPS leads to enhanced production of NT3 and other neural growth factors [20]. Neurotrophins are synthesized as pro-proteins, and they are spliced to generate the mature forms, either before or after its exocytosis. We detected two bands of NT3 approximately 30 kDa and 65 kDa, respectively, while the molecular size of the mature NT3 monomer is around 15 kDa, and the pro-form of NT3 is approximately 30 kDa. However, it is known that NT3 is present in cells as homodimer or heterodimer with a brain-derived neurotrophic factor [21,22]. Dimers of mature NT3 is approximately 30 kDa, and those pro-form should be approximately 60 kDa. Thus, NT3 in human AVICs is in dimeric forms.

The effect of LPS on NT3 levels in human AVICs is dependent on TLR4 function since blocking TLR4 with a specific neutralizing antibody abolished the upregulation of NT3 production by LPS. Several endogenous factors, such as soluble biglycan and oxidized low-density lipoprotein (ox-LDL) can function as damage-associated molecular patterns (DAMPs) to activate TLR4 in human AVICs [26,27]. These DAMPs may be involved in the upregulation of NT3 production in diseased aortic valves that display higher TLR4 levels in AVICs [28].

The Akt and ERK1/2 pathways play a role in mediating osteogenic responses in human AVICs [18,23]. In this study, we found that both Akt and Erk1/2 pathways are involved in mediating TLR4-induced NT3 production in human AVICs. The activation of Akt and ERK1/2 in human AVICs appears to be the direct result of TLR4 signaling since phosphorylation of Akt and ERK1/2 occurred rapidly after the stimulation of TLR4. It is noteworthy that the inhibition of either Akt or ERK1/2 abrogated the induction of NT3 production. It appears that the concomitant activation of these two pathways is required for the upregulation of NT3 in human AVICs through TLR4 and that the activation of one of these two pathways is insufficient to enhance NT3 production.

The inflammatory, fibrogenic, and osteogenic activities in aortic valves seem to interact [5,9], and, more specifically, the stimulation of human AVICs with pro-inflammatory stimuli enhances both inflammatory and osteogenic activities [9,28]. In the present study, we found that the stimulation of TLR4 with pro-inflammatory stimulus LPS promotes cell proliferation and elevates fibrogenic activity in human AVICs. Thus, TLR4 signaling in human AVICs is also pro-fibrogenic. It seems that the stimulation of TLR4 in human AVICs upregulates NT3 production to promote cell proliferation since cell proliferation is accompanied by increased levels of NT3, and the neutralization of NT3 markedly attenuates TLR4-induced AVIC proliferation. Interestingly, the proliferative rate in cells treated by neutralization of NT3 remains slightly higher, although this difference, compared to the baseline level, is insignificant. It is likely that the remaining increment in proliferation rate is caused by cytokines since several pro-inflammatory cytokines are capable of inducing cell proliferation [29]. Alternatively, the moderate upregulation of bone morphogenetic protein-2 and/or transforming growth factor-beta 1 by TLR4 may account for the remaining increment in proliferation rate as both of these two growth factors are proliferative in vascular cells [30]. Nevertheless, NT3 plays a major role in mediating the AVIC proliferation induced by the stimulation of TLR4.

The effects of NT3 are mainly mediated by the Trk and p75 neurotrophin receptors [14]. NT3 could bind to TrkA, TrkB, and TrkC to activate these three Trk receptors [31]. Our previous study demonstrated that Trk has a major role in mediating NT3-induced responses in human AVICs [18,19]. In this study, we confirm that the inhibition of the Trk receptor with pan inhibitor K252a essentially abrogated the NT3-induced cell proliferation in human AVICs. Similarly, the stimulation of TLR4 with LPS failed to induce human AVIC proliferation in the presence of this Trk inhibitor or the NT3 neutralizing antibody. In view of the fact that neutralizing NT3 and blocking Trk have comparable effects on TLR4-induced human AVIC proliferation, we propose that the NT3-Trk cascade mediates the proliferative effect of TLR4 stimulation on human AVICs. However, it is unclear which type of Trk receptor mediates the effect of NT3 on AVICs. Future studies are needed to analyze the expression of Trk receptors in human AVICs and to evaluate the relative role of each type of the receptors.

CAVD pathobiology involves excess production and deposition of ECM proteins, particularly collagens [32]. Collagen deposition in the ECM provides a scaffold upon which nodular calcification can occur [33]. Thus, excess collagen production and deposition not only exacerbate valvular fibrosis, but also promote valvular calcification. To assess the effect of TLR4 stimulation on human AVIC fibrogenic activity, we analyzed cellular levels of collagens and MMPs, and probed for collagen deposition. We found that the stimulation of TLR4 with LPS upregulates the expression of collagen III and MMP-9 in human AVICs. More importantly, it results in greater collagen deposition when cells are exposed to LPS for a prolonged period. Thus, the stimulation of TLR4 in human AVICs elevates cellular fibrogenic activity. The effect of TLR4 activation on fibrogenic activity is mediated by the NT3-Trk cascade since NT3 neutralization and the Trk blockade each markedly reduces the effect of TLR4 stimulation on AVIC fibrogenic activity. It is well known that myofibroblasts have higher fibrogenic activity than fibroblasts [34]. It remains unknown from this study whether NT3 induces myofibroblastic transition in human AVICs to elevate their fibrogenic activity. Our previous studies found that soluble matrilin 2 upregulates α-smooth muscle action (α-SMA) levels and elevates the fibrogenic activity in human AVICs [35], while ox-LDL elevates human AVIC fibrogenic activity without the upregulation of cellular levels of α-SMA [36]. It appears that the induction of myofibroblastic transition in human AVICs is stimulus-specific and that either cell activation or cell phenotype transition may elevate human AVIC fibrogenic activity.

Greater collagen deposition is viewed as in vitro fibrosis [37]. The finding that the stimulation of TLR4 in human AVICs promotes collagen deposition through a mechanism involving the NT3-Trk cascade is significant for improving the understanding of mechanisms underlying aortic valve fibrosis. However, the translation of the in vitro finding to in vivo conditions should be done with caution.

MMP-9 is involved in maintaining fibroblast viability and promoting the production of pro-inflammatory factors, including IL-1β, IL-6, IL-8, and TNF-α in fibroblasts [38]. In addition, MMP-9 overproduction in AVICs is believed to promote valvular ECM remodeling [39]. However, the relative activity of MMP is modulated by their specific inhibitors, the tissue inhibitors of MMPs (TIMPs). TLR4 stimulation may also modulate the level TIMPs in human AVICs. Therefore, the significance of MMP-9 upregulation by TLR4 stimulation is unclear. Further study will assess the impact of TLR4 stimulation on the expression of MMPs and their inhibitors in human AVICs in the context of normal and pathological conditions.

This study has some limitations. First, the sample size is small. We repeated the experiments five times using distinct cell isolates from different donor valves and found a satisfactory reproducibility of the data with acceptable variability. However, the data would be more comprehensive if cells from more donor valves were analyzed. Second, aortic valves from explanted hearts affected by cardiomyopathy may have subtle changes in valvular structure. In this regard, valvular ECM structure has been found to be altered in the valves of hearts from patients with ischemic heart disease or dilated cardiomyopathy [40]. In this study, we isolated AVICs from the aortic valves of explanted hearts of heart transplant recipients with cardiomyopathy. Although our previous study confirmed that AVICs isolated from such valves display osteogenic responses to pro-inflammatory stimulation in the same manner as AVICs isolated from valves of normal hearts [28], an influence of heart disease on the in vitro fibrogenic responses of AVICs to pro-inflammatory stimulation could not be excluded.

## 4. Materials and Methods

### 4.1. Material

Polyclonal antibodies against collagen III (catalog ab83829), NT3 (catalog ab53685), and the NT3-neutralizing antibody (catalog ab6203) were purchased from Abcam (Cambridge, MA, USA). Recombinant human NT3 was obtained from US Biological (San Antonio, TX, USA). The TLR4-neutralizing antibody (catalog HTA125) was purchased from Santa Cruz Biotechnology (Santa Cruz, CA, USA). The thymidine analog 5-bromo-2′-deoxyuridine (BrdU) cell proliferation kit and antibodies against phosphorylated Akt (catalog 9271), total Akt (catalog 9272), phosphorylated ERK1/2 (catalog 9010), and total ERK1/2 (catalog 4695), MMP-9 (catalog 3852), and Glyceraldehyde 3-phosphate dehydrogenase (GAPDH) (catalog 2118, clone number 14C10) were purchased from Cell Signaling, Inc. (Beverly, MA, USA). The Trk inhibitor K252a was obtained from Calbiochem (Darmstadt, Germany). The Akt inhibitor MK2206 and ERK1/2 inhibitor PD98059 were purchased from Selleckchem (Houston, TX, USA). The Cell Counting Kit (CCK)-8 was purchased from Enzo Life Sciences International (Plymouth Meeting, PA, USA). Medium 199 was purchased from Lonza (Walkersville, MD, USA). Picro-Sirius Red and other reagents were purchased from Sigma-Aldrich Chemical (St Louis, MO, USA).

### 4.2. Cell Isolation and Culture

Normal aortic valves were collected from explanted hearts of patients (6 males and 1 female; age 58.7 ± 3.4 years) who had cardiomyopathy and were undergoing heart transplantation at the University of Colorado Hospital. The valve leaflets from explanted hearts were thin, and no abnormality was revealed by hematoxylin and eosin staining. In a previous study, we confirmed that AVICs isolated from such “normal’ valves and those from true normal valves (from donor hearts for transplant) have comparable responses to pro-inflammatory stimulation [28].

This study was approved by the University of Colorado Institution Review Board (IRB Protocol 08-0280; approval Date 01/05/2016). All subjects gave their informed consent for the use of their aortic valves for this study. The investigations were carried out following the rule of the Declaration of Helsinki of 1975, revised in 2013.

AVICs were isolated and cultured using a previously described method [10]. Briefly, valve leaflets were subjected to an initial digestion with a higher concentration of collagenase (2.5 mg/mL) to remove endothelial cells. Then, the remaining tissue was treated with a lower concentration of collagenase (0.8 mg/mL) to free interstitial cells. Cells were collected by centrifugation and were cultured in an M199 growth medium containing penicillin G, streptomycin, amphotericin B, and 10% fetal bovine serum. AVIC isolates obtained using this modified protocol lack endothelial cells, as verified by the negative von Willebrand factor staining [10]. One isolate was obtained from a donor aortic valve and was used as an independent sample. Thus, 6 normal AVIC isolates were used for this study. Cells of passage 3 to 6 were used for the experiments.

When grown to 50–60% confluence, AVICs were stimulated with LPS (0.20 µg/mL) or NT3 (0.10 µg/mL) to determine the effect of the reagents on human AVIC proliferation. The cell proliferation was analyzed by BrdU ELISA and CCK-8 assays When grown to 80–90% confluence, AVICs were stimulated with LPS for various time periods. When needed, cells were pretreated with pharmacological reagents, including Trk inhibitor (K252a, 0.20 µM), Akt inhibitor (MK2206, 5.0 µM), ERK1/2 inhibitor (PD98059, 25 µM), and TLR4-neutralizing antibody (10 µg/mL) or NT3-nuetralizing antibody (10 µg/mL) 1 h prior to the addition of LPS.

### 4.3. Immunoblotting

Immunoblotting was applied to analyze NT3, collagen III, MMP-9, phosphorylated Akt, total Akt, phosphorylated ERK1/2, total ERK1/2, and GAPDH in cell lysate. Cells were lysed in a sample buffer (100 mmol/L Tris-HCl, pH 6.8, 2% sodium dodecyl sulfate (SDS), 0.02% bromophenol blue, and 10% glycerol). The protein samples were separated on gradient (4–20%) mini-gels and were transferred onto nitrocellulose membranes (Bio-Rad Laboratories, Hercules, CA, USA). The membranes were blocked with a 5% skimmed milk solution for 1 h at room temperature. The blocked membranes were incubated with a primary antibody. After washing with phosphate-buffered saline (PBS) containing 0.05% Tween 20, the membranes were incubated with a peroxidase-linked secondary antibody specific to the primary antibody. After additional washes, the membranes were incubated with enhanced chemiluminescence reagents and exposed on x-ray films. Image J (Wayne Rasband, National Institutes of Health, Bethesda, MD, USA) was used to assess the density of bands.

### 4.4. BrdU Proliferation Assay

Cell proliferation was analyzed using a BrdU kit according to manufacturer’s protocol. AVICs were plated in 96-well plates at a density of 50–60% confluence. The experiments were performed in triplicate and began by replacing the medium with fresh medium containing different concentrations of the compounds under investigation. The cells were incubated for 3 days with the compounds. Twenty-four hours prior the end of the experiment, BrdU labeling solution was added to the wells. The amount of BrdU in the DNA was subsequently determined by fixation of the cells and incubation with an anti-BrdU-POD antibody followed by colorimetric analysis using a Bio-Rad model/680 microplate reader at 450 nm. The results were expressed as a % of control value.

### 4.5. CCK-8 Assay

A CCK-8 assay was applied to confirm cell proliferation. AVICs were plated in 96-well plates at a density of 50–60% confluence. The experiments were performed in triplicate and began by replacing the medium with a fresh medium containing different concentrations of the compounds under investigation. The cells were incubated for 3 days with the compounds. After the incubation was complete, the metabolic activity was determined using the CCK-8 assay according to the manufacturer’s protocol. In brief, cells were incubated with CCK-8 labeling reagent at 37 °C for 4 h. Formazan dye formation was evaluated by scanning with a multi-well spectrophotometer at 450 nm. The results are expressed as a % of control value.

### 4.6. Picro-Sirius Red Staining

Picro-Sirius Red (PSR) staining specifically identifies collagens, and is a useful method for the assessment of fibrogenic activity in cultured cells [37]. The cells were treated with methanol overnight at −20 °C. Following washes with PBS, the cells were incubated in 0.1% PSR for 4 h. Following rinses with 0.1% acetic acid, plates were air-dried and examined by microscopy. For quantification of PSR staining, stained cells were treated with 100 µL of 0.1 M sodium hydroxide for 2 h to elute the stain. The optical density was determined using a spectrophotometer (BioTek Instruments, Inc., Winusky, VT, USA) at 540 nm. The results are expressed as a % of change.

### 4.7. Statistical Analysis

The data are presented as mean ± SE. Statistical analysis was performed using StatView software (Abacus Concepts, Calabasas, CA). The ANOVA with the post hoc Bonferroni/Dunn test was used to analyze differences between experimental groups, and a *t*-test was applied to analyze data comparisons between control and LPS-treated AVICs. A nonparametric Mann-Whitney U test was performed to confirm the difference of two group comparisons. For multiple group comparisons, a nonparametric Kruskal-Wallis test was performed to confirm the differences. Statistical significance was defined as *p* ≤ 0.05.

## 5. Conclusions

The novel findings of this study include: (1) the activation of innate immunoreceptor TLR4 elevates the fibrogenic activities in human AVICs, which is associated with the up-regulation of the cellular expression of neurotrophin NT3, (2) the TLR4 ligand induces NT3 expression in human AVICs through the concomitant activation of the AKT and ERK1/2 pathways, (3) NT3 plays a major role in mediating the pro-fibrogenic effect of TLR4 activation and (4) NT3 modulates the fibrogenic activities in human valvular cells through the Trk receptor. These findings demonstrate that NT3 is a novel pro-fibrogenic factor in human AVICs and indicate that the upregulation of NT3 production by endogenous TLR4 activators may contribute to the mechanism underlying aortic valve fibrosis. The findings of the present study not only advance the understanding of the inflammatory mechanism of CAVD progression but also suggest therapeutic targets for suppression of aortic valve fibrosis.

## Figures and Tables

**Figure 1 ijms-21-01276-f001:**
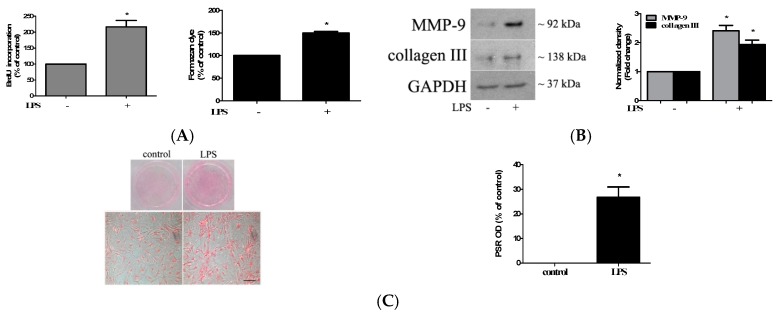
Stimulation of TLR4 upregulates aortic valve interstitial cell (AVIC) fibrogenic activity. (**A**) Human AVICs were treated with lipopolysaccharide (LPS) (0.20 µg/mL) for 72 h. The bromodeoxyuridine/5-bromo-2′-deoxyuridine (BrdU) and cell counting kit-8 (CCK-8) assays show that LPS induces higher cell proliferation. (**B**) The representative immunoblots and densitometric data show that collagen III and metallopeptidases-9 (MMP-9) are increased in cells exposed to LPS for 72 h. (**C**) AVICs were treated with LPS for 28 days. The representative images of Picro-Sirius Red staining (scale bar = 150 µm) and spectrophotometric data show that collagen deposition is markedly increased by TLR4 stimulation. All data are presented as mean ± SE of five experiments using different cell isolates from distinct valves. * *p* < 0.05 vs. untreated control.

**Figure 2 ijms-21-01276-f002:**
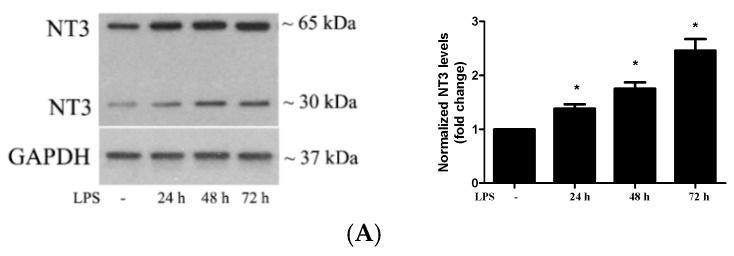
Stimulation of TLR4 upregulates neurotrophin 3 (NT3) production in human AVICs. (**A**) AVICs were treated with LPS (0.20 µg/mL) for 24, 48, or 72 h. The representative immunoblots and densitometric data show that LPS increases cellular NT3 levels in a time-dependent fashion. (**B**) The AVICs were treated with LPS (0.20 µg/mL) for 72 h in the presence or absence of the TLR4-neutralizing antibody (10 μg/mL). The representative immunoblots and densitometric data show that the neutralization of TLR4 markedly reduces LPS-induced NT3 production. All data are presented as mean ± SE of five experiments using different cell isolates from distinct valves. * *p* < 0.05 vs. untreated control; # *p* < 0.05 vs. LPS alone; and & *p* < 0.05 vs. LPS + Immunoglobulin G (IgG).

**Figure 3 ijms-21-01276-f003:**
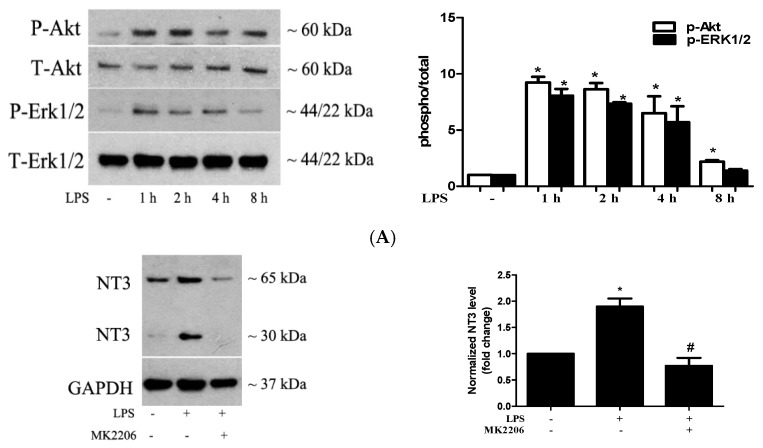
Akt and extracellular signal-regulated protein kinases 1 and 2 (ERK1/2) are involved in the mechanism by which TLR4 induces NT3 production in human AVICs. (**A**) The AVICs were exposed to LPS for 1 to 8 h. The representative immunoblots and densitometric data show that LPS induces the rapid phosphorylation of Akt and ERK1/2. (**B**) The AVICs were treated with LPS for 72 h in the presence or absence of an Akt inhibitor (MK2206, 5.0 µM) or an ERK1/2 inhibitor (PD98059, 25 µM). The representative immunoblots and densitometric data show that the inhibition of Akt or ERK1/2 reduces LPS-induced NT3 production. All data are presented as mean ± SE of five experiments using different cell isolates from distinct valves. * *p* < 0.05 vs. untreated control; # *p* < 0.05 vs. LPS alone; and & *p* < 0.05 vs. LPS + dimethyl sulfoxide (DMSO).

**Figure 4 ijms-21-01276-f004:**
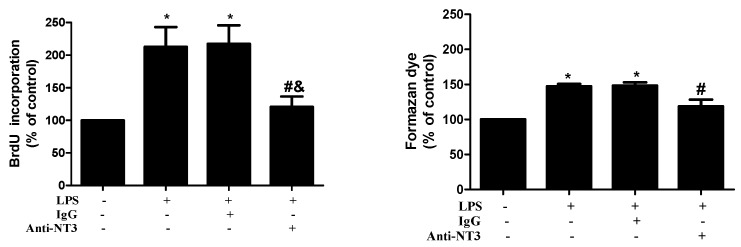
NT3 mediates human AVIC proliferation induced by the stimulation of TLR4. The AVICs were treated with LPS (0.20 µg/mL) for 72 h in the presence or absence of the NT3-neutralizing antibody (10 μg/mL). The BrdU and CCK-8 assays show that the neutralization of NT3 markedly reduces LPS-induced cell proliferation. The data are presented as mean ± SE of five experiments using different cell isolates from distinct valves. * *p* < 0.05 vs. untreated control; # *p* < 0.05 vs. LPS alone; and & *p* < 0.05 vs. LPS + IgG.

**Figure 5 ijms-21-01276-f005:**
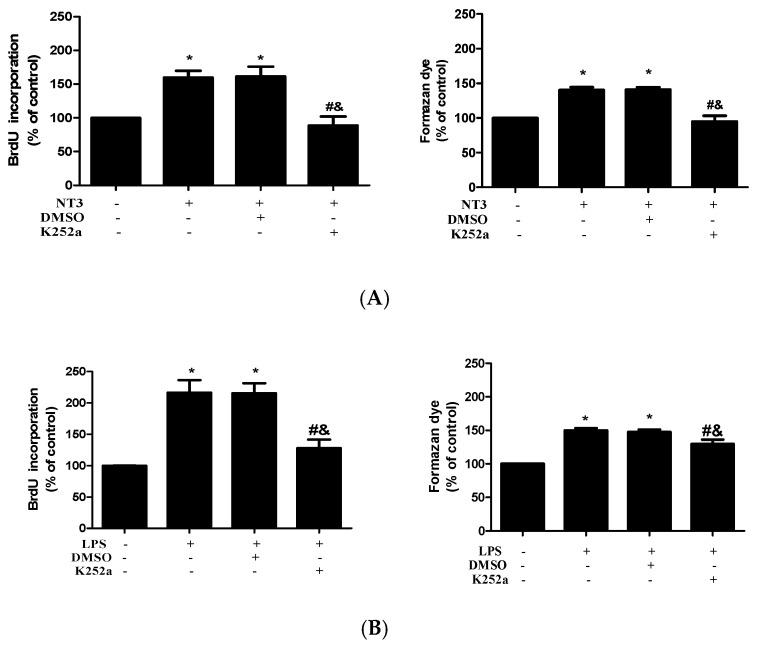
Trk mediates human AVIC proliferation induced by NT3 and LPS. (**A**) The AVICs were treated with NT3 (0.10 µg/mL) for 72 h in the presence or absence of a Trk inhibitor (K252a, 0.20 μM). The BrdU and CCK-8 assays show that the inhibition of Trk abrogates NT3-induced AVIC proliferation. * *p* < 0.05 vs. untreated control; # *p* < 0.05 vs. NT3 alone; and & *p* < 0.05 vs. NT3+DMSO. (**B**) The human AVICs were treated with LPS (0.20 µg/mL) for 72 h in the presence or absence of a Trk inhibitor (K252a, 0.20 μM). The BrdU and CCK-8 assays show that blocking Trk markedly reduces LPS-induced AVIC proliferation. All data are presented as mean ± SE of six experiments using different cell isolates from distinct valves. * *p* < 0.05 vs. untreated control; # *p* < 0.05 vs. LPS alone; and & *p* < 0.05 vs. LPS + DMSO.

**Figure 6 ijms-21-01276-f006:**
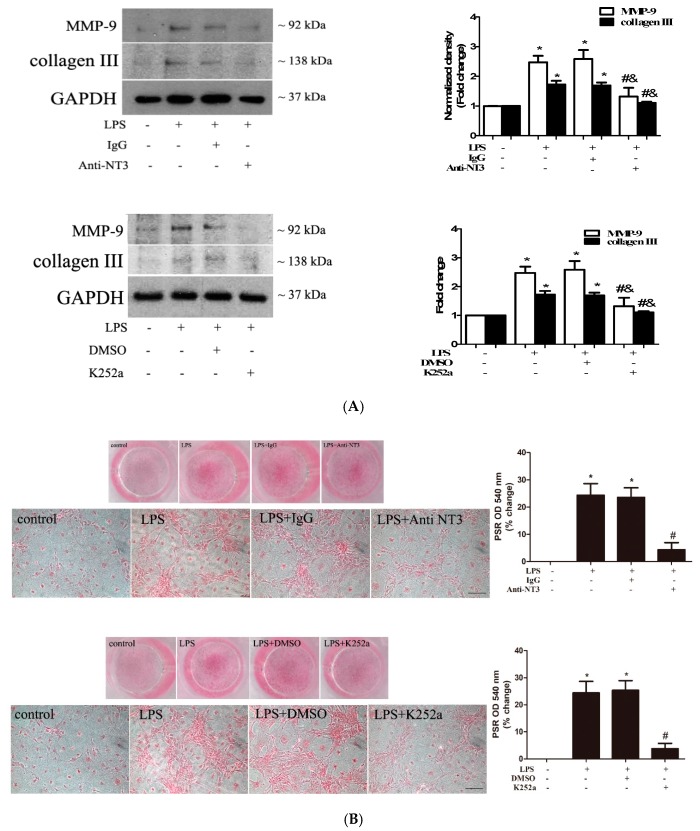
NT3-Trk cascade is involved in mediating LPS-induced extracellular matrix (ECM) protein expression and collagen deposition in human AVICs. (**A**) The AVICs were treated with LPS for 72 h in the presence or absence of an NT3-neutralizing antibody (10 µg/mL) or Trk inhibitor (K252a, 0.20 µM). The representative immunoblots and densitometric data show that the neutralization of NT3 or the inhibition of Trk suppresses the LPS-induced expression of collagen III and MMP-9. (**B**) The AVICs were treated with LPS for 28 days in the presence or absence of the NT3-neutralizing antibody (10 µg/mL) or Trk inhibitor (K252a, 0.20 µM). The representative images of Picro-Sirius Red staining (scale bar = 150 µm) and spectrophotometric data show that collagen deposition induced by LPS is markedly reduced by the neutralization of NT3 or the inhibition of Trk. * *p* < 0.05 vs. untreated control; # *p* < 0.05 vs. LPS alone; and & *p* < 0.05 vs. LPS + IgG or LPS + DMSO.

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
