# Peer review of "TLR4 Stimulation Promotes Human AVIC Fibrogenic Activity through Upregulation of Neurotrophin 3 Production"

_ijms, 2020, doi:10.3390/ijms21041276_

Round 1
Reviewer 1 Report
The authors analyzed the hypothesis that stimulation of TLR4 in human AVICs upregulates NT3 production to elevate cellular fibrogenic activity.
Presented study had a small number of examined valves. Despite limitations of this study, presented manuscript might be appropriate for the Journal, but a few more issues should be considered before acceptance
Prestend topic seem to be well explored and descibed. What is new in this study? What is clinical importance and possible clinical imlications? Why mean and SE (not standard deviation) was used? Authors examined explanted hearts of patients with cardiomyopathy and were undergoing heart transplantation. There is posssible bias elated to unknow factos related to basic disease. Why hearts of completly healthy patients were used? For example heart of victims of suicides? Small sample size might limit results of the study.
Author Response
Novelty and significance of this study
We apologize for not stating clearly the novelty of findings and their significance. This is the first study demonstrating that activation of innate immunoreceptor TLR4 elevates the fibrogenic activities in human AVICs, which is associated with up-regulation of neurotrophin NT3 expression in human valvular cells. Mechanistic experiments revealed that TLR4 ligand induces NT3 expression in valvular cells through the AKT and ERK1/2 pathways. More importantly, novel findings demonstrate that NT3 plays a major role in mediating the pro-fibrogenic effect of TLR4 activation and that NT3 modulates the fibrogenic activities in human valvular cells through Trk receptor.
The pathobiology of calcific aortic valve disease involves valvular fibrosis and calcification. Severe calcific aortic valve disease requires costly aortic valve replacement, and moderate aortic valve thickening due to fibrosis can increase the risk of other cardiovascular events, such as ischemic heart disease and left ventricular dysfunction. Currently, pharmacological intervention for calcific aortic valve disease progression is lacking due to the gaps in our knowledge of the mechanism underlying the pathobiology of this disease. The findings of this study provide new insights into the inflammatory mechanisms of calcific aortic valve disease progression and indicate novel targets for suppression of aortic valve fibrosis.
To emphasize the novelty and significance of this study, we summarized main novel findings and commented on their significance in the Discussion (page 7) and Conclusion (page 11).
The reason for using mean±SEM
Standard deviation (SD) calculates the dispersion or the variability of a population/dataset around the mean of that particular population. Thus, SD is a measure of the variability within a population/dataset. Standard error of the mean (SEM) is a measure that quantifies how far your sample is likely to be from the mean of the population. SEM reflects the average variability of repeated experiments. Since SEM quantifies the precision of the mean obtained from repeated experiments, we elected to use mean±SEM to present the experimental data.
Aortic valves from hearts affected by cardiomyopathy and small sample size
Thank you for the comments.
While the human valve leaflets that we used in this study were from explanted hearts of transplant patients with cardiomyopathy, they have normal morphology. In addition, we have confirmed that AVICs isolated from such “normal’ valves and those from true normal valves (from donor hearts for transplant) have comparable responses to pro-inflammatory stimulation (Yang X et al., Pro-osteogenic phenotype of human aortic valve interstitial cells is associated with higher levels of Toll-like receptors 2 and 4 and enhanced expression of bone morphogenetic protein 2. J Am Coll Cardiol 2009, 53:491-500). We included this information in the Materials and Methods section of the revised manuscript (page 10). In addition, we acknowledged the potential effect of factors related to heart disease on valvular structure and commented this in the Discussion section as one of the limitation of this study (page 9).
With regard to the sample size, we agree with the reviewer that a larger sample size would make the data more comprehensive. However, we repeated the experiments 5 times using distinct cell isolates from different donor valves and found a satisfactory reproducibility of the data with acceptable variability. Therefore, analysis of additional samples with multiple experiments (with each cell isolate from a different donor valve as a sample) would not provide additional information. Nevertheless, we acknowledged in the Discussion section (page 9) that a small sample size is one of the limitations of this study.
Reviewer 2 Report
In this manuscript, Yao et al. investigate the role of TLR4 in the aortic valve interstitial cell (AVIC) fribrogenic activity through the upregulation of NT3 protein. The authors claim that TLR4 activation through lipopolyssacharide leads to activation of Akt and MAPK signaling pathways that are required to increase NT3 levels, activation of Trk receptors and AVIC fibrogenic activity. Although the results seem to be of some interest, the only novelty is that TLR4 activation increases NT3, since the roles of NT3 in the proliferation and collagen production by AVIC and in the osteogenic responses have been reported previously (Yao et al., BBA 2015 and Yao et al., Am J Physiol Cel Physiol 2017). However, I have a major concern. The antibody utilized to identify NT3 protein detects bands at different sizes than the 15 kDa band expected for mature NT3 (Fig. 1 and Supplemental material). This issue should be properly addressed.
Major points:
1- Figure 2. Size indicated for NT3 is around 65 kDa, when mature NT3 size is around 15 kDa. I am not confident that the band showed in Fig 2A and in the supplemental material really corresponds to NT3. Since NT3 expression is based in a unique antibody, it will be convenient to confirm the results with an additional, independent NT3 antibody. The same antibody has been previously used for Fig 1 (Yao et al., Am J Physiol Cel Physiol 2017) indicating again a 65 kDa band, what it is a little bit strange.
2- Akt and MAPK required individually to upregulate NT3. What about both together? Is there any additive effect?
3- The authors state that Trk receptors mediate NT3 functions. Which Trk receptor, TrkA, TrkB or TrkC, is involved in NT3 actions? In the previous paper by the authors (Yao et al., Am J Physiol Cel Physiol 2017), they assess using TrkA Fc, TrkB Fc and TrkC Fc bodies to quench NT3, without any clear conclusion. This does not indicate which receptor is responsible for NT3 actions in AVICs. Specific antibodies should be used to detect expression of TrkA, TrkB or TrkC in AVICs. K252a inhibits all three Trk receptors and, therefore, it does not be useful to identify which one is express/involved in NT3 actions.
4- The first two bars presented in the graphs corresponding to Fig 4 (LPS – and +) seem to be the same as the ones in Fig 1A. If so, it should be stated in the text.
Minor points:
Page 6 lines 191-193. The sentence is incorrect, a verb is missing.
Author Response
We want to take this opportunity to thank this reviewer for thoroughly reviewing our manuscript. I feel that the reviewer’s constructive comments are very helpful for improving the quality of this manuscript.
Figure 2. Size indicated for NT3 is around 65 kDa, when mature NT3 size is around 15 kDa. I am not confident that the band showed in Fig 2A and in the supplemental material really corresponds to NT3. Since NT3 expression is based in a unique antibody, it will be convenient to confirm the results with an additional, independent NT3 antibody.The reviewer was correct about the molecular size of mature NT3 monomer. However, the pro-form of NT3 is bigger, approximately 30 kDa. Literature also documented that NT3 is present in cells as homodimer or heterodimer with brian-derived neurotrophic factor (Kolbeck R et al., Characterisation of neurotrophin dimers and monomers. Eur J Biochem. 1994, 225:995-1003; Robinson RC et al., Structure of the brain-derived neurotrophic factor/neurotrophin 3 heterodimer. Biochemistry. 1995, 34:4139-4146). Dimers of mature NT3 is approximately 30 kDa, and those pro-form should be approximately 60 kDa. We apologize for not describing this in the manuscript.
We constantly detect two specific bands, approximately 30 and 65 kDa in cell lysate of human aortic valve cells using non-reducing immunoblotting. This is also true for tissue homogenate of rat spinal cord, and different antibodies (Abcam catalog ab6203 and Abcam catalog ab53685) give the same result. As the 65 kDa band is stronger and sharper, we presented only that band in the figures. Following your critiques about the size of NT3 protein band, we replaced the NT3 gels in Figures 2 and 3 with the uncropped gels unloaded with the initial submission. Thus, the gels in these two figures now show the two bands, representing dimers of mature and pro-form of NT3 protein. For clarification, we described the protein band sizes in detail in the Results section (page 3) and commented on the difference of band size in the Discussion section (page 8).
Akt and MAPK required individually to upregulate NT3. What about both together? Is there any additive effect?
Thank you for the question. Our results show that inhibition of either Akt or ERK1/2 abolishes the up-regulation of TN3 expression, indicating that simultaneous activation of these two pathways is required for the up-regulation. Inhibition of both pathways is unlikely to have an additive effect since inhibition one already voids the effect. Nevertheless, we made a comment on this in the Discussion section (page 8).
The authors state that Trk receptors mediate NT3 functions. Which Trk receptor, TrkA, TrkB or TrkC, is involved in NT3 actions? Specific antibodies should be used to detect expression of TrkA, TrkB or TrkC in AVICs. K252a inhibits all three Trk receptors and, therefore, it does not be useful to identify which one is express/involved in NT3 actions.
Thank you for the comment and the excellent suggestion. We changed the phrase “Trk receptors” to “Trk receptor”. We made a speculation in the Discussion section and stated that future studies are needed to analyze the expression of Trk receptors in human aortic valve cells and to evaluate the relative role of each isoform of the receptors (page 8).
The first two bars presented in the graphs corresponding to Fig 4 (LPS – and +) seem to be the same as the ones in Fig 1A. If so, it should be stated in the text.
We made the statement in page 5.
Page 6 lines 191-193. The sentence is incorrect. A verb is missing.
We apologize for the oversight. We added the missing verb and double-checked the entire manuscript.
Round 2
Reviewer 1 Report
Manuscript could be accepted in current form
Author Response
Thank you so much!
Reviewer 2 Report
The authors have addressed my previous concerns.
There is just one minor issue. In page 8 line 289 the word isoform is incorrect since TrkA, B and C neurotrophin receptors are encoded by different genes and therefore they are not isoforms.
Author Response
Following your recommendation, we have changed “which Trk isoform” in line 292 to “which type of Trk receptor” and changed “each isoform of the receptors” in line 294 to “each type of the receptors”. In this R2 version of the manuscript, the statements read as the following:
However, it is unclear which type of Trk receptor mediates the effect of NT3 on AVICs. Future studies are needed to analyze the expression of Trk receptors in human AVICs and to evaluate the relative role of each type of the receptors.